# Efficient Strategies for Better Imbalance Image Segmentation

**Dawood Chanti**[1]                                          DAWOOD.ALCHANTI@LS2N.FR
**Vanessa Gonzalez Duque**[1]                      VANESSA.GONZALEZDUQUE@LS2N.FR
**Marion Crouzier**[2]                                MARION.CROUZIER@UNIV-NANTES.FR
**Antoine Nordez**[2]                                 ANTOINE.NORDEZ@UNIV-NANTES.FR
**Lilian Lacourpaille**[2]                         LILIAN.LACOURPAILLE@UNIV-NANTES.FR
**Diana Mateus**[1]                                         DIANA.MATEUS@LS2N.FR

[1] *École Centrale de Nantes, Laboratoire des Sciences du Numérique de Nantes LS2N, UMR CNRS 6004 Nantes, France.* [2] *Université de Nantes, Laboratoire "Movement - Interactions - Performance", MIP, EA 4334, F-44000 Nantes, France.*

**Editors:** Under Review for MIDL 2021

## Abstract

We propose a strategy that encourages filter reuse to decrease the total number of learned parameters and to enable training on small dataset efficiently. We also highlight on one of our recent publication (Al Chanti et al., 2021), which handles foreground/background class imbalance by learning adaptively how to penalize False Positives and False Negative pixels, resulting in a faster convergence and better performance. We validate our method on low limb muscle segmentation using volumetric ultrasound.

**Keywords:** Parametric Tversky loss, filters reuse, 3D ultrasound, muscle segmentation.

## 1. Introduction

One major challenge in medical image segmentation is properly training Convolutional Neural Networks (CNNs) when annotated data (i) is limited, which reduces the network generalization performances, (ii) suffers from class imbalance, which makes network predictions biased towards the non-organ class and (iii) has complex image texture and resolution such as those coming from Ultrasound (US) data (Duque et al., 2020), which makes feature extraction and representation harder. *In this paper*, we focus on the class imbalance solution proposed in our recent publication (Al Chanti et al., 2021) and based on the *parametric Tversky loss function*. Additionally, we argue that dealing with limited amount of annotated data and complex image textures can be addressed by learning a deep network with a *reduced number of learned parameters* by reusing learned filters and dilate them at different rates, which are parameter free yet extract multi-scale and contextual features.

## 2. Method and Materials

**Shared dilated filters:** We design a 3D fully convolutional network for volumetric data with an architecture adapted from the 3D Seg-Net-FS (Al Chanti et al., 2021) but with important modifications. Our network is composed of a 3D encoder that includes five 3D convolutional layers. Each layer is constructed as represented in Fig. 1 where learned filters are reused with dilation to capture image context and not learned as in (Chen et al., 2017).

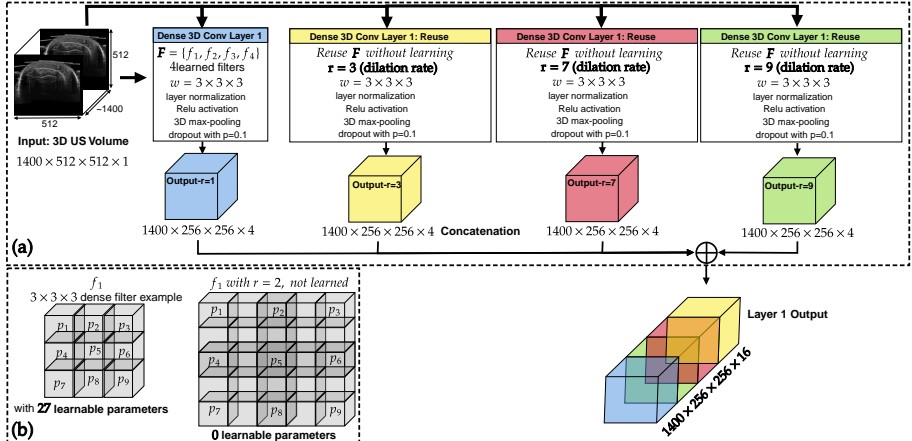

Figure 1: Extracting multi-scale and contextual features (Fig. (a)) with less learnable parameters. Similar to Atrous convolution, we adapted similar setting for dilating the filters (Fig. b). However, our approach encourage filters reuse to enhance pattern detection in the image context and to learn minimal number of parameters. The total number of learned parameters for the first encoder layer are 120 instead of 480 in the typical setting with learned dilated filters (Chen et al., 2017).

Each layer is composed consecutively of $4, 8, 16, 32, 64$ learned filters. Each layer produces $16, 32, 64, 128, 256$ feature maps. The 3D decoder uses similar setup as the 3D encoder with the difference that the 3D max-pooling operation is replaced by 3D up-sampling operation. The final layer from the 3D Decoder is passed to an output layer with softmax activation to produce the volumetric segmentation maps. The total number of parameters resulted from this model is only 1,031,606 while if we replaced our reused layers with typical atrous convolution with learned filters, the total number of parameters becomes 4,123,826.

**Learned Tversky loss for class imbalance:** let $\hat{\boldsymbol{y}}$ and $\boldsymbol{y}$ be the set of predicted and ground truth binary labels respectively. A typical Dice loss function weights false positive (FP) and false negative (FN) voxels equally, which causes the learning process to get trapped in local minima of the loss function, yielding predictions that are strongly biased towards the background. As a result, the foreground region is often partially detected. Hence, weighting FNs more than FPs is crucial. We propose to parameterize the Tversky Index (Eq. (1)) to include two learnable parameters $\alpha$ and $\beta$ that control the magnitude of penalties for FPs and FNs instead of tuning them manually. Our loss function is $1 - \text{TI}$.

$$\text{TI}(\hat{\boldsymbol{y}}, \boldsymbol{y}, \alpha, \beta) = \frac{\sum_i^N \hat{\boldsymbol{y}}_{0i} \boldsymbol{y}_{0i}}{\sum_i^N \hat{\boldsymbol{y}}_{0i} \boldsymbol{y}_{0i} + \alpha \sum_i^N \hat{\boldsymbol{y}}_{0i} \boldsymbol{y}_{1i} + \beta \sum_i^N \hat{\boldsymbol{y}}_{1i} \boldsymbol{y}_{0i}} \tag{1}$$

$\hat{\boldsymbol{y}}_{0i}$ is the probability of voxel $i$ being a foreground of a target muscle and $\hat{\boldsymbol{y}}_{1i}$ is the probability of voxel $i$ being a background. The same applies to $\boldsymbol{y}_{0i}$ and $\boldsymbol{y}_{1i}$ respectively. Typically, we start with $\alpha$ and $\beta$ equal to 0.5, which reduces Eq. (1) to dice loss. Then, $\alpha$ and $\beta$ gradually change their values, such that they always sum up to 1. In order to guarantee that $\alpha + \beta = 1$, we apply a softmax function to generate a probability distribution.

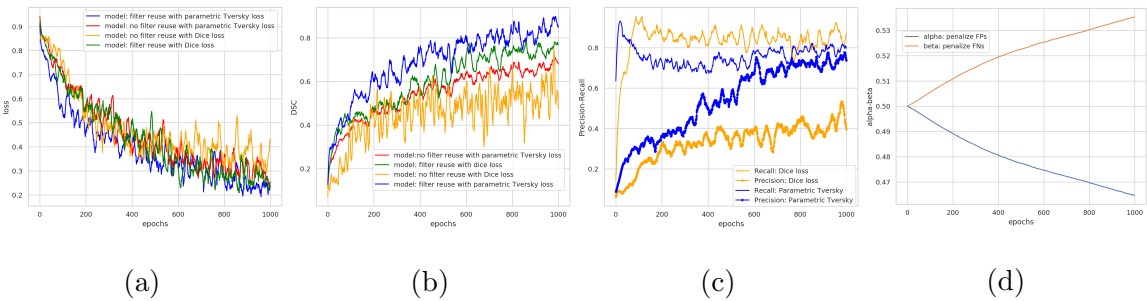

Figure 2: Models behaviour over 1000 epochs over the validation set. (a) learning curve. (b) DSC. (c) precision-recall curve. (d) Tversky learned penalization parameters.

## 3. Experimental Results

**Data:** A 3D US recordings of low limb muscle for 44 participants are used. Each recording filling a voxel grid of $512 \times 512 \times 1443 \pm (49 \times 38 \times 207)$, and having the annotations of the Gastrocnemius Medialis (GM), the Gastrocnemius Lateralis (GL), and the Soleus (SOL) muscles. Data description available in (Duque et al., 2020) and (Al Chanti et al., 2021). **Experimental Setting:** we measure the model behaviour with filter reuse strategy and with learned dilated filters along with dice and parametric Tversky losses. **Results:** Fig. 2(a) shows that a faster convergence is achieved when using parametric Tversky loss function. Fig. 2(b) shows that a better Dice Similarity Coefficient (DSC) score is obtained when using filter reuse strategy, especially when trained using parametric Tversky loss. Fig. 2(c) shows the advantages of parametric Tversky loss as it permit a trade-off between the precision and the recall to provide a better segmentation maps. It can be observed that dice loss totally fail as it has low precision and high recall. Fig. 2(d) shows the gradual update of the values alpha and beta during the learning process. We reported 81% abd 76% of DSC scores over the test set using a model with filter reuse strategy with tversky loss and with dice loss respectively. Then we reported 68% and 49% of DSC scores using a model with learned dilated filters with Tversky loss and with dice loss respectively.

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
