# OpenReview forum: "Efficient Strategies for Better Imbalance Image Segmentation"
_MIDL.io/2021/Conference/Short — Submitted to MIDL 2021_

### Official Review · Reviewer_wLSK · 2021-05-04

**Confidence:** 4
**Final Rating:** 1

**Summary:**

In this paper, the authors present a method for image segmentation specifically designed to deal with small, imbalanced datasets of noisy data, like e.g. ultrasound images. Thus, they present a CNN architecture with few parameters by integrating the reuse of filters throughout different scales and propose a loss function with trainable parameters to avoid bias arising from class imbalance. The authors provide a proof-of-concept study on a segmentation task in 3D ultrasound images.


**Strengths:**

The work aims on tackling multiple problems in medical imaging at once, while all of them are of high relevance. The idea to let the network adjust its own loss function towards generalization is neat. The results are promising, both for the novel loss function as well as the filter reuse approach.


**Weaknesses:**

Unfortunately, the strength of the manuscript is also its weakness: The manuscript is methodologically crowded and hence is missing a clear goal or red line. Dealing with imbalanced data and minimizing the number of parameters to train robustly on small data seem to be quite distinct problems from my point of view. Hence, the presentation and evaluation of two methods is pressed into one short paper. This would not be problem if there was no lack of significant details on the methods. Unfortunately, exactly that is the case. Specifically, it is not clear how the parameters \alpha and \beta were adjusted. As the authors claim to apply softmax activation, I assume that there is some dense layer predicting them, but the architecture remains unknown. Additionally, the data split is not described and hence, the size of the training set remains unclear. There is no discussion of the results or the methods, the paper ends with presenting quantitative results without drawing conclusions. At least a short discussion/conclusion is necessary.
In 2, the authors claim that they „propose to parameterize the Tversky Index“, which is at least prone to misconception as the Tversky Index is by parameterized by \alpha and \beta per definition. The authors propose to learn these parameters, but this is a different claim.
In addition, the paper is not very well-written. Fig. 2 is way too small to be beneficial, and there are a lot of typos, specifically in section 3 (see detailed comments below).


**Deanonymize Review:**

no

**Detailed Comments:**

Fig. 2 should be large enough to be readable. As mentioned above, there are several typos:
-page 2, line 8: Start capital after colon
-page 3, line 1: Delete „A“
-page 3, line 5: start capital after colon
-page 3, line 10: permits
-page 3, line 11: delete „a“
-page 3, line 12: fails
-page 3, line 13: write \alpha and \beta as greek letters
-page 3, line 13: and


**Justification Of The Rating:**

Even though I am sure that the presented method as well as the results are very interesting, I cannot vote for accepting this manuscript in its current form. It feels like pressing all developed methods from a journal paper into a short paper, sacrificing a clear goal as well as the red line. Several core points of the method as well as the evaluation remain unclear and the figures showing the results are barely assessable. Additionally, the text is not in an adequate form.
Therefor, I do not think that the manuscript matches the high standards of MIDL. One again, this decision is not due to the developed method or the results but due to the presentation of them.


**Paper Type:**

methodological development

**Special Issue:**

no

---

### Official Review · Reviewer_q7FX · 2021-05-05

**Confidence:** 4
**Final Rating:** 3

**Summary:**

This paper proposes a tuned 3d-network architecture for 3d segmentation, to reduce its number of learnable parameters; which in turn reduces overfitting and improve learning.
Moreover, the authors propose to learn the $\alpha$ and $\beta$ from the Tversky index, which seems to improve performances even further.

**Strengths:**

- the validation is performed on a difficult modality in general (ultrasounds)
- the authors report both numbers and curves
- the authors compare the different combinations of losses/network optimizations

**Weaknesses:**

My main issue is that the authors do not compare to a Tversky loss with fixed $\alpha$ and $\beta$. As such, it is not clear if the improvement over dice loss comes from the parametrization itself.

Note: finding automagically the optimal $\alpha$ and $\beta$ is a good contribution, if tuning by hand proves cumbersome. But at least a quickly tuning by hand should be added as a baseline

**Deanonymize Review:**

no

**Justification Of The Rating:**

The paper shows some improvements over the baseline with their proposed method, but some issue remain with its baseline comparison (no 'static' Tversky loss). But I think that it is an interesting contribution for the MIDL community.

**Paper Type:**

methodological development

**Special Issue:**

no

---

### Meta-Review · Area_Chair_TzwF · 2021-05-07

**Recommendation:** Reject
**Confidence:** 5

**Metareview:**

The AC agrees that the paper is missing focus and should at least provide some form of discussion of the results.

---

### Decision · Program_Chairs · 2021-05-11

Reject